

# Technical note: a Weighing Forest Floor Grid-Lysimeter

Heinke Paulsen[1], Markus Weiler[1]

[1]Chair of Hydrology, Albert-Ludwigs-Universität Freiburg, Freiburg, 79098, Germany

*Correspondence to*: Heinke Paulsen (Heinke.paulsen@hydrology.uni-freiburg.de)

**Abstract.** The forest floor (FF) is dominated by plant litter and its decomposition products, thereby it differs significantly from the mineral soil. Because of its wider range of pore sizes and overall high porosity, it has a large capacity to retain water and thus plays an important role in redistributing water to the mineral soil beneath. Until now most studies have focused on the behaviour of the organic layer when wetted and dried in a laboratory setting. Alternatively, field fresh samples were collected to determine the water storage potential. We present a novel low-cost grid-lysimeter designed specifically for the FF, but also

suitable for other organic soil layers. It can continuously measure all water balance components of the FF. The lysimeter detects precipitation with an accuracy of 0.03 mm outperforming most rain gauges. The developed setup allows for further customization of in-situ water quality measurements. This technical note describes the setup of the lysimeter and presents performance metrics from laboratory results and initial field data.

## 1 Introduction

The forest floor (FF) is hydrologically highly relevant but is only partially explored (Floriancic et al., 2023). This organic layer covering forest soils offers a huge potential to store and retain water and also plays an important role in the redistribution of water to deeper soil horizons. Consequently, it affects infiltration patterns and might also induce runoff generation. For example, FF interception has a large influence on the water balance as it alters water amount available for soil infiltration and runoff (Guevara-Escobar et al., 2007). Typical canopy ecosystem services like water infiltration, filtration, soil erosion control

can also be attributed to the FF.

FF can also reduce the amount of water reaching the soil, how much depends on the physical features of the FF and rainfall characteristics. Several studies revealed, that the storage capacity of FF is proportional to mass and thickness (Putuhena and Cordery, 1996; Zagyvai-Kiss et al., 2019), while the capacity of broadleaves to intercept water is greater than the one of needle litter due to a higher surface area to weight ratio (Li et al., 2020; Walsh and Voigt, 1977; Zhao et al., 2022). The studies

investigated the behaviour of the organic layer under wetting and drying conditions in laboratory settings, utilizing rainfall simulator experiments, or collected field samples post-rainfall events, subsequently oven-drying them to assess water storage potential.

Currently, there is limited data regarding the potential dynamics of the FFs contribution to a forests water cycle. First approaches were undertaken by Gerrits et al. (2007), who developed a simple weighing device, similar to a lysimeter, to

directly measure evaporation from the FF in the field. Floriancic et al. (2023) conducted a comparative analysis of soil moisture





dynamics with and without FF coverage. They found that FF interception can reach up to 20-50 % of the total precipitation (Gerrits et al., 2007) and retains water for up to 2 days and longer (Floriancic et al., 2023). The results indicate that neglecting FF interception in modelling, alongside canopy interception or treating it as a static percentage, may lead to significant overestimations of recharge and transpiration rates in water balance assessments.

Our objective was to develop a device for investigating the dynamics of the water balance components of a FF. Weighted lysimeters are a well-established method to measure water fluxes in agricultural contexts (von Unold and Fank, 2008). They allow for direct assessment of all water balance components, including various forms of precipitation (like rain, dew, and rime), drainage, evapotranspiration, and storage (Reth et al., 2021). Since lysimeters provide measurement at ground level their results are close to the true precipitation as the precipitation measurements are unaffected by wind or precipitation

intensity, unlike traditional rain gauges (Schnepper et al., 2022). Based on this we developed the cost-effective Forest Floor Grid-Lysimeter (FFGL), a weighted lysimeter with limited depth (covering only the organic layer plus the uppermost mineral soil) facilitating the deployment of multiple devices across various locations. In contrast to existing devices the FFGL is adaptable to slope, which allows for surface alignment even on steep hillslopes. With its smaller surface area, a fourth of the device developed by Gerrits et al. (2007), and the separation into four grids and individual percolation measurements, it allows

for exploring the heterogeneity of water fluxes at an even smaller spatial scale. With the combination of 3D printed parts and a customized microcontroller board, the costs could be reduced allowing comparative observations among many sites. Additionally, the FFGL allows for the later adaption to measure water quality parameters like electrical conductivity or concentrations of dissolved organic carbon (DOC).

## 2 Methods

### 2.1 General setup

The Forest Floor Grid-Lysimeter (FFGL) was designed to generate data with high temporal resolution across multiple locations and varying spatial scales. Therefore, a low-cost setup to install multiple lysimeters at several study sites was necessary. To explore the small-scale heterogeneity of infiltration patterns, we partitioned the lysimeter into four grids, facilitating a typical grid lysimeter approach that enables the observation of outflow from each grid independently. Fig. 1 illustrates the FFGL,

comprising three main parts: a weighted container (1) containing the FF, a frame (2) supporting the measurement equipment and securing the system in the soil, and the control unit (3). The FFGL covers an area of 25 cm x100 cm. In our case it is filled with forest floor, but could contain variable fillings, depending on the application (e.g. plants, deadwood, etc.). The FFGL quantifies the main water fluxes of the FF: precipitation (throughfall when beneath the canopy), evaporation (evapotranspiration in case of ground vegetation), interception of the FF (storage) and percolation to the mineral soil (generally

equivalent to infiltration into the soil).

(1)   The container constructed from stainless steel, has a height of 25 cm and a maximum volume of 62.5 l. The bottom plate of the box is inclined, resulting in the formation of four grids, each draining through distinct openings. Additionally, these



four grids are divided by plastic partition walls that can be easily cut for easy adjustment to different slopes in the field. On top of these dividing walls lies a perforated metal sheet wrapped with gardening fleece to retain the forest floor
material. The container is placed on the load cells, which are affixed to the frame, without further securing measures.

(2) The frame supports the measurement equipment. The container rests on four load cells (LCs) mounted to the corners of the frame. Thereby the weight of the lysimeter can be continuously measured. Water is collected by a funnel into a measurement unit (MU) beneath each grid hole. This unit controls the intensity of water flow onto the tipping buckets (TBs), thereby enhancing measurement accuracy. Additionally, the MU allows for later adaption to also measure water
quality parameters like temperature, electrical conductivity (EC) etc. The TBs measure the amount of draining water. Some lysimeters are equipped with a water collector at the bottom for subsequent laboratory analysis, while others are designed to allow water to infiltrate freely.

(3) The control unit will be explained in the next section.

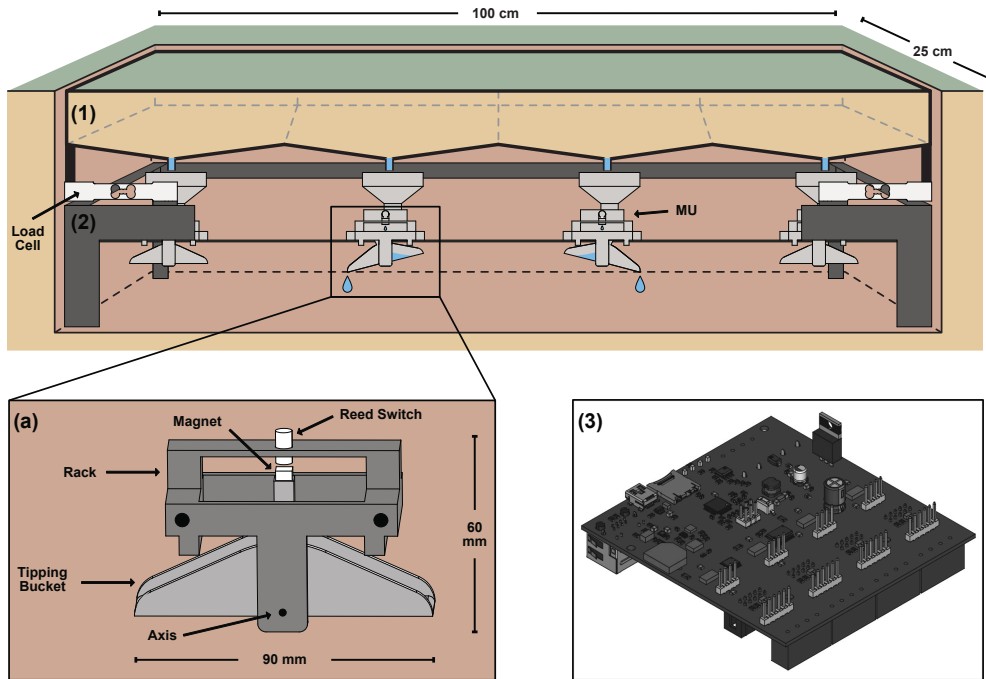

**Figure 1: Schematic drawing of the lysimeter setup. Containing the (1) container and (2) frame holding the load cells and (a) tipping buckets. (3) shows the customized microcontroller board.**



## 2.2 Control Unit

### 2.2.1 Hardware

The electronics of the lysimeter was designed to meet user-friendly and economic criteria. Therefore, we used cost-effective

hardware and a microcontroller compatible with the Arduino IDE. The objective was to achieve high energy efficiency, since the lysimeters are located far from a power source and must autonomously measure continuously for at least one month. The computing unit must efficiently perform floating-point calculations to directly process the measurements. We opted for commercially available micro SD cards for data storage. A real-time clock (RTC) was incorporated to associate the measurements with a timestamp. To enable the use of the FFGL as an SDI-12 sensor, we incorporated this feature into the

board. System-relevant parameters such as the SDI-12 bus address are stored on the board.

The connectivity of the circuit board includes the following ports:

For lysimeter measurements:

- 4x load cell inputs
- 4x temperature measurements
- 4x EC analog input
- 4x pulse input for reed contacts of the tipping buckets

For control and data output:

- 1x SDI-12
- 1x USB
- 1x SD-Slot
- 1x RTC

### 2.2.2 Microcontroller, IDE & Software

To accelerate the development process, we decided to use the Arduino IDE and its supported hardware. This platform is broadly used in environmental monitoring and supports various microcontrollers (MCUs). For initial tests, we used an older

AVR Mega 2560 controller because it has a wide range of analog and digital inputs and outputs. However, this controller has disadvantages such as an outdated 8-bit architecture, high power consumption, and a slow system clock of only 16 MHz. After researching various microcontrollers supported by the Arduino platform, the ATSAMD21G18A-48 proved to be the most suitable and cost-effective option. The microcontroller, produced by AVR/Microchip, incorporates a M0+ processor along with adequate flash memory and RAM. The advantage of this microcontroller is its high compatibility with old Arduino

libraries while offering many new features with its Cortex architecture. This MCU operates at 3.3 V with a clock speed of 48 MHz and is 32-bit, allowing for fast and energy-efficient evaluation of measured values. The ADC has a significantly higher resolution at 12-bit compared to the old MEGA2560's 10-bit resolution. The controller is used in some official Arduino boards like the Arduino Zero or the industrial series Arduino MKR. However, since these boards do not fulfil our specifications, we have created our own board that accommodates the controller along with the necessary peripherals. To program our board, we



simply flash the bootloader of the MKR Zero and program it like an MKR Zero but with the advantages of our own hardware peripherals. The board was created using Altium Designer layout and schematic software.

The microcontroller was programmed using Arduino IDE due to its user-friendly interface, ease of learning, open-source nature, and compatibility with various hardware components. We developed a program that reads the sensors in defined timesteps. In combination with event-based programming and depending on the electrical power supply, data can be collected

more often (i.e. every minute) during rain events and less often (i.e. every 10 minutes) during drier periods. This contributes to energy conservation. Program code, STL files for printing and board design can be found on https://github.com/HeinkePaulsen/Forest-Floor-Grid-Lysimeter.

### 2.3 Load Cells

For the mass measurement of the FFGL container we used four load cells (LCs) H10A from BOSCHE with a single weighing

range of 15 kg together with HX711 loadcell amplifiers. These LCs operate with strain gauges. They transform forces, in our case pressure, into an electrical output that can be measured and standardized. The change in resistance of the strain gauges can be quantified as voltage. This change in voltage is proportional to the amount of force applied to the cell, thus the mass of the FF can be calculated from the LC output.

Each LC was calibrated individually to attain high accuracy by applying a known mass and averaging five analogue readings.

To obtain the calibration factor that needs to be put into the program, the known weight is divided by this analog value. We conducted precision tests to assess the accuracy of mass measurements for the lysimeter container. Therefore, we placed masses ranging from 0 and 10,000 g in the center of the lysimeter. Additionally, we made some tests where we placed the mass in the four different grids of the lysimeter.

### 2.4 Tipping Bucket

The tipping bucket (TB) consists of a 3D printed bucket and frame. These parts are connected with a metal pin. Additional parts include a magnet and a reed switch. The TB can easily accommodate larger or smaller tipping volumes by replacing it with differently-scaled TBs without further changes of the overall set-up. The principle of a tipping bucket is straightforward. The water accumulates in one chamber of the bucket until it reaches the weight to induce tipping subsequently allowing the other chamber to fill. The tipping is recorded when the magnet closes the reed switch during tipping.

We used either Bambulab P1P or a Prusa MK2 for our 3D printing process. To guarantee high stability and long-term robustness in the environment we used PETG (polyethylene terephthalate glycol) filament, which delivers a significant chemical resistance, durability and formability and shows no interference with water quality measurements. Since we want to achieve a very small tipping volume, it is important to use the feature "ironing" for 3D-printing the tipping buckets. This ensures a smooth surface and reduces adhesion of water to the material. To avoid back-bouncing of the TB against the frame

whilst tipping - which generates erroneous double tips - small shock absorbers made from hot glue were added at the frame contacts.



Various experiments were conducted for the static calibration of the tipping buckets. Continuous dripping with a pipette, 20 g and 50 g. As typical for TBs, the tip volume depends on the water intensity flowing into the TB. But since in our lysimeter the water has to passage through the forest floor, we assume dropping water rather than high flow rates onto the TB. Also, the MU

is created to have a rather small outlet which restricts the intensity to a specific level, thereby enhancing the system's accuracy.

## 2.5 Filling of Lysimeters and Location/Positioning in the field

The FFGL boxes were filled during the summer of 2023 (August to October). The objective was to fill the lysimeters with FF-material as undisturbed as possible and to recreate the soil layers to avoid big rocks and roots redirecting water fluxes. We decided to fill the boxes with the upper 15 cm of material from the surface accommodating litter fall in autumn. First, we sliced

off the FF layer in one piece and put it aside. The soil below was taken into boxes in 5 cm layers to a depth of 15 cm for subsequent reconstruction. The perforated metal sheet in the container was aligned to the actual surface slope in the field using the separating plastic walls. On top, we laid a thin garden fleece onto which we reconstructed the soil layer by layer within the box. The undisturbed O-layer was placed on top (Fig. 2).

To fit the lysimeter frame and box into the location where we removed the FF, the excavation pit was enlarged to 50 cm depth

and slightly wider, ensuring the surface of the lysimeter aligns with the surrounding FF. The excavation hole walls were reinforced with wooden plates to prevent contact between the lysimeter walls and the surrounding soil. Then the frame was inserted and aligned for balance. Then the electronics was connected and the box inserted. Lifting the container onto the frame from the top allows for easy access for cleaning purposes and maintenance. A lysimeter container used for testing in the laboratory was equally filled and then transported to the laboratory for functionality tests.



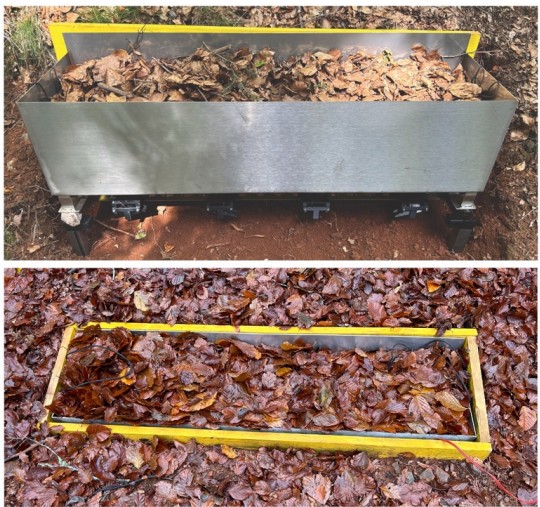


**Figure 2: Top: Lysimeter container on the frame, bottom: fully installed FFGL aligned to the location.**

### 2.6 Water Balance Calculation

The measurements of a weighing lysimeter can be used to determine the water balance of the FF for each observed time step. The amount of water percolating to the deeper soil $D$ can be determined by multiplying the number of tipping bucket tips $n$

with the tipping volume. With an average tipping volume of 2.1 cm³ and the area covered by each lysimeter grid we reach a resolution of 0.03 mm for the draining water.

$$D \ = \ n * \frac{2.1\,\text{cm}^3}{625\,\text{cm}^2}, \tag{1}$$

The load cells continuously measure the storage $S$ and, consequently, the mass of the lysimeter. If there is no drainage and the

weight change is negative, we know that this change is due to evaporation $E$ from the FF.

$$\text{If } D \ = \ 0 \ \& \ \Delta S \ < \ 0,$$
$$E \ = \ \Delta S, \tag{2}$$

A positive storage change is a signal for precipitation in our case canopy throughfall ($P_{TF}$). This precipitation/canopy throughfall can be calculated as:

$$P_{TF} \ = \ E + D + \Delta S. \tag{3}$$





Under the assumption there is no evaporation during a precipitation event.

**3 Results**

**3.1 Mass quantification with Load Cells**

To evaluate the accuracy of the mass quantification, we loaded the container with known masses ranging between zero to 10 kg in 500 g steps and conducted 20 measurements. The mass measurement accuracy achieved is 0.3%. A second test was conducted to confirm this, in which a fixed mass of 1000 g was placed at various positions within the container, specifically

at the center of each grid and at the overall center of the container (Fig. 3a). This test yields in an overall precision and standard deviation of 2.0 g, corresponding to 0.008 mm of precipitation. Figure 3a also shows that the position of the mass in the lysimeter has a influence on the measured mass. Therefore, the mass of the filled container should be distributed quite evenly on all four LCs to ensure accurate results.





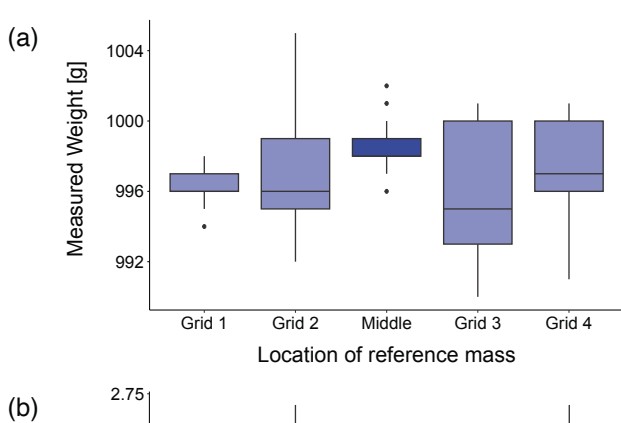

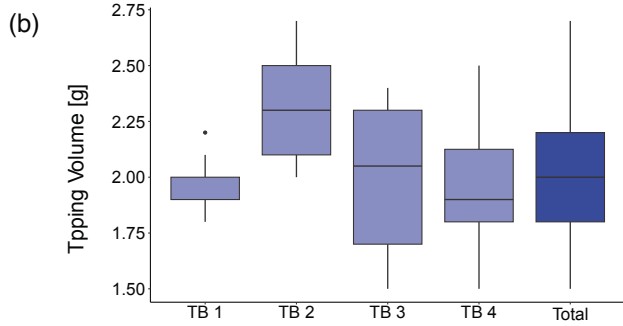

**Figure 3: (a) Weighing precision of load cells, depending on the location of the reference mass in the container and (b) tipping volume of four single tipping buckets below one lysimeter.**

**3.2 Quantification of percolating water with Tipping Buckets**

The calibration of the TBs yields in a mean tipping volume of 2.1 ml (Fig. 3b), corresponding to a drainage resolution of 0.03 mm per grid (25 x 25 cm). The standard deviation lies at 0.3 ml equivalent to 0.0048 mm. The error is 2%. We also show here the differences among different tipping buckets. For the four tested TBs the mean tipping volumes lie between 1.9 and 2.3 ml, equivalent to 0.031 mm and 0.037 mm. The observed minor differences are probably caused by the printing accuracy of the 3D printer. The resolution is comparable and acceptable when compared to other TBs mounted in rain gauges. For example the commercially available HOBO Raingauge Data Logger RG3 (https://www.veldshop.nl/en/hobo-rain-gauge-data-logger-rg3-m.html) has a resolution of 0.1 mm with an error of 1% and the tipping bucket raingauge by ecoTech (https://www.ecotech.de/en/product/tipping_bucket_rain_gauge_va) also has a resolution of 0.1 mm. Lysimeters employed in





Ruth et al. (2018) quantify draining water with a 50-mL tipping bucket, yielding a resolution of approximately 0.016 mm and with a Mini-Lysimeter the resolution was 0.5 g, corresponding to a water column of approximately 0.007 mm.

### 3.3 Irrigation experiment

We performed an irrigation experiment in the laboratory to test the performance and accuracy of the whole lysimeter setup under controlled conditions. The results are presented in Table 1. The mass of the empty box is 13.5 kg. We filled it with 21 kg of FF consisting of mineral soil and organic layers. We irrigated a total amount of 4875 ml, corresponding to 19.5 mm of precipitation. We separated the irrigation into three artificial events.

**Table 1: Results of the irrigation experiment.**

| Event | Time | Applied P [mm] | Measured P [mm] | Accuracy [%] | D [mm] |
|---|---|---|---|---|---|
| 1 | 11-12:30 | 11.4 | 11.7 | -2,6 | 1.6 |
| 2 | 12:30-15:00 | 3.8 | 3.8 | 0 | 1.8 |
| 3 | 15-16:30 | 4,3 | 3.7 | 14.0 | 2.4 |
| **Total** | **11-16:30** | **19.5** | **19.2** | **1.5** | **5.8** |


In Fig. 4 we plotted the amount of measured precipitation and draining water, as well as the cumulative fluxes during the irrigation in 10-minute increments. Due to the dried-out material, drainage started after 16 minutes and 8.6 mm of water applied. For the second and third event the drainage occured much faster (3 minutes, with 0.5 & 0.2 mm water applied) due to the higher initial water content and also a larger part of the irrigated water drained from the box.

Comparing the cumulative fluxes, a major part of the irrigated water 70.1% remained stored in the FF and did not percolate. The overall statistics reveal that we applied 19.5 mm water, but the lysimeter only measured 19.2 mm, resulting in a 1.5% divergence from the applied amount. This difference could be explained by evaporation during the irrigation which cannot be measured due to the measuring principle of the lysimeter and also water missing the lysimeter due to fumigation caused by the nozzle of the irrigation device. In the following ten hours after the last TB tip, we could observe a further decline in mass

without any further water drainage. Based on this, we can deduce that 1.8 mm of water was evaporated from the lysimeter container, accounting for 13% of the irrigated water.



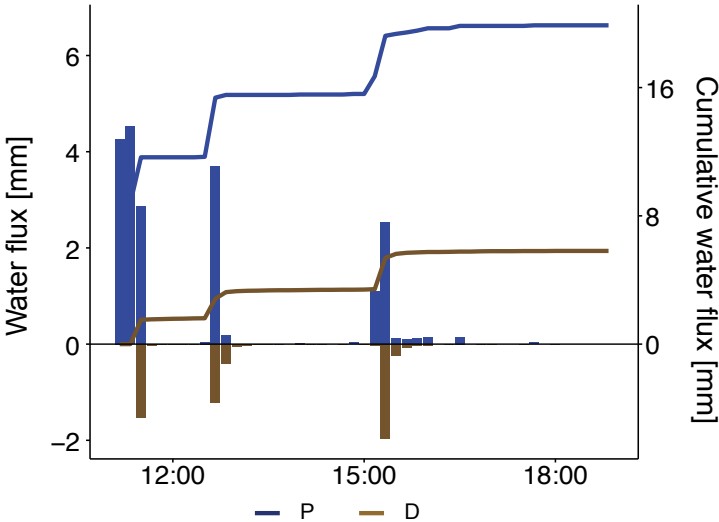

**Figure 4: Results of the irrigation experiment; water fluxes (bars) as well as accumulated fluxes (lines).**

### 3.4 Initial field results

To demonstrate the performance of the FFGL in field conditions, we provide the results of two lysimeters located in the Conventwald, an experimental site in a mixed forest stand of the University of Freiburg in the Black Forest over a 10 day period. The site is located 20 km east of Freiburg at 840 m a.s.l. and has a mean annual precipitation of 1749 mm, more site characteristics can be found in Rinderer et al. (2021). We positioned the FFGLs under two spruce trees - one under the crown edge (SCE) and one in the crown middle (SCM) area - 1.5 m apart from each other. In Fig.5 we compare the lysimeter data to

the above canopy precipitation. The total amount of above canopy rainfall (P) during the observed time period was 101.4 mm, we split it into four separate rainfall events (Table 2). With the lysimeters at the two locations SCE and SCM we measured 81.1 mm and 62.1 mm in total throughfall ($P_{TF}$), respectively. That equals to a canopy interception loss of 20.0% and 38.8% for the two locations, which is comparable to other studies. For example, Gerrits et al. (2010) observed losses of 18% in broadleaved stand while Andreasen et al. (2023) observed 35 % loss in broadleaf and 44 % in coniferous forest stands.


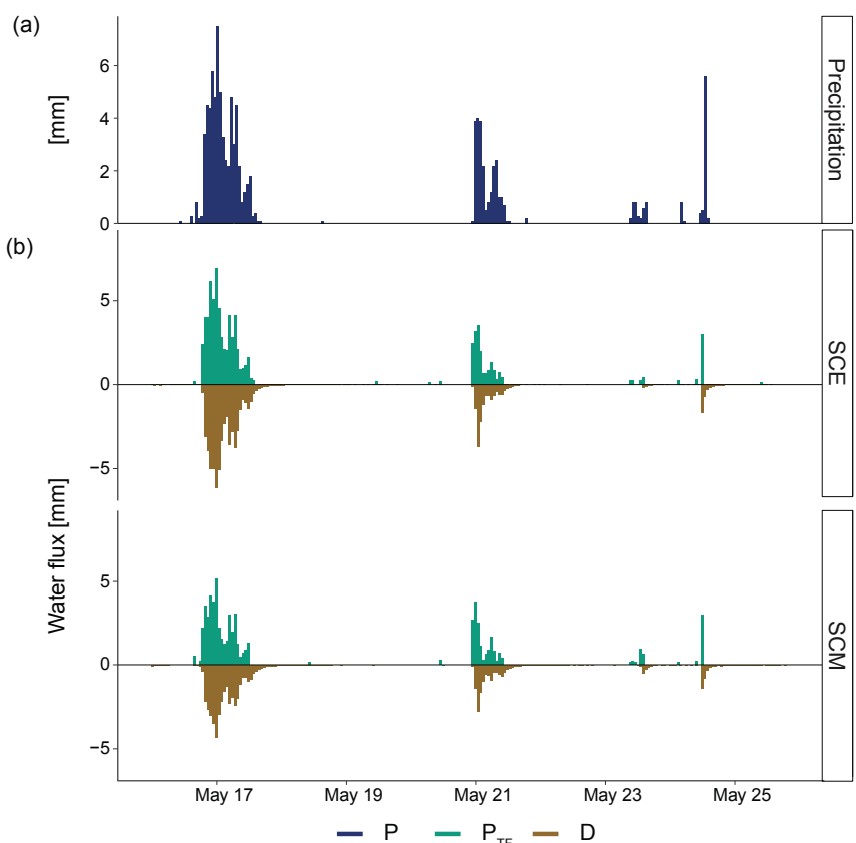

**Figure 5: (a) Precipitation (P) measured above canopy and (b) Canopy throughfall ($P_{TF}$) and Drainage (D) measured by the two lysimeters below the crown edge and middle of a spruce tree.**

Looking at the individual events (Table 2) it becomes evident, that the fraction of rainfall reaching the ground is highly

dependent on both the event and the position of the lysimeter. The fraction of the $P_{TF}$ reaching a lysimeter fluctuates between

35 and 90% during the analyzed time period due to the shape of the crown and varying rainfall conditions (wind speed, drop size, wind direction). Smaller rainfall events cause a smaller partition of canopy throughfall, resulting in less percolation to deeper soil horizons.




**Table 2: Observed Water fluxes for the two lysimeters during the 10 day period, sectioned into four events.**

| | | SCE | | | | SCM | | | 250 |
| 16.5.-26.5. | P | $P_{TF}$ | $P_{TF}/P$ [%] | D | $D/P_{TF}$ [%] | $P_{TF}$ | $P_{TF}/P$ [%] | D | $D/P_{TF}$ [%] |
|---|---|---|---|---|---|---|---|---|---|
| Event 1 | 65.8 | 59.1 | 89.8 | 58.3 | 98.6 | 41.1 | 62.5 | 40.0 | 97.3 |
| Event 2 | 24.3 | 17.7 | 72.8 | 14.6 | 82.5 | 15.6 | 64.2 | 12.7 | 81.4 |
| Event 3 | 3.7 | 1.3 | 35.1 | 0.6 | 46.2 | 2.1 | 56.8 | 1.5 | 71.4 255 |
| Event 4 | 7.5 | 3.8 | 50.7 | 3.6 | 94.7 | 3.3 | 44.0 | 3.4 | 103.0 |
| **Total** | **101.4** | **81.1** | **80.0** | **77.2** | **95.2** | **62.1** | **61.2** | **57.7** | **92.9** |

Drainage occurs with a delay to the start of each precipitation event, showing that it needs time for the water to flow through
the FF. The cumulative fluxes in Fig. 6 show that there is a divergence in both time and amount. Only 95.2 and 92.9 % of $P_{TF}$
become D, respectively. Not all of the P reaching the ground does infiltrate to deeper soil horizons but is stored in the FF and
might later evaporate. In the study period only 3.3 and 1.5 mm evaporated, respectively, accounting for 4.1 and 2.4 % of $P_{TF}$.
This part could be higher in drier and warmer seasons when potential evaporation rates are higher.

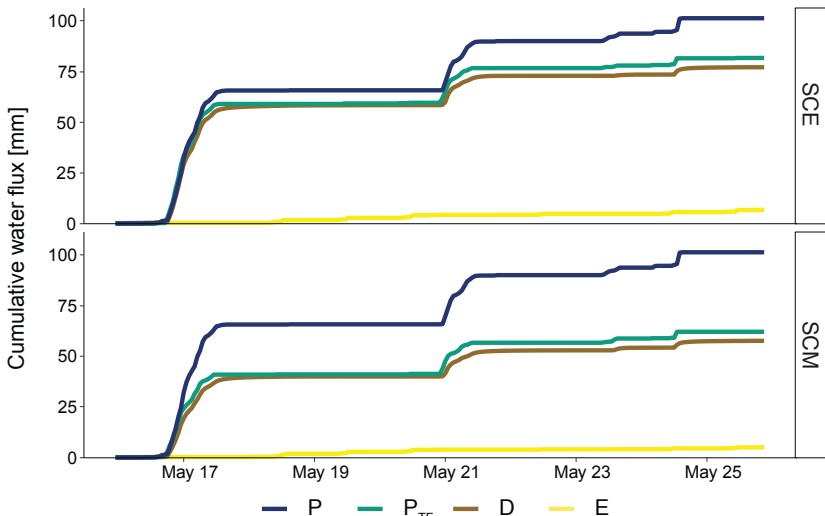

**Figure 6: Cumulative water fluxes including Precipitation (P) measured on the tower, canopy throughfall ($P_{TF}$), Drainage (D) and
Evaporation (E) measured by the two lysimeters.**



$P_{TF}$ accounts up for a major part of P, depending on the position of the lysimeter under the trees. In addition, due to the spatial heterogeneity in throughfall, the FF further influences infiltration pathways. This becomes clear when looking at the percentage of drainage through the various lysimeter outflows. It becomes clear that some of the grids have much higher amounts of

drainage than others (Fig. 7). For example, during the study period, the outermost right grid (TB 4) of SCM received over 50% of the draining water while TB 3 had only 5%. One could state that this could also be due to spatial heterogeneity in $P_{TF}$, but since the mass gain on all four LCs was similar, we conclude that this deviation in infiltration must be due to rerouting pathways in the FF itself.

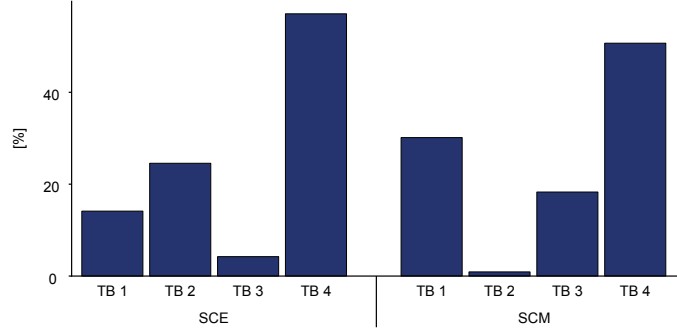

**Figure 7: Percental share of infiltration through the different grids of the two lysimeters.**

Figure 8 compares the soil moisture data collected by the SMT100, measured 50 cm upslope to the lysimeters at 5 and 15 cm depth to the water stored in the lysimeter container. An increase in stored water of 5 mm compares to an increase in soil moisture of 10 %. Soil moisture and storage are higher at the crown edge compared to the crown middle. The storage dynamics measured in the lysimeters is more pronounced compared to the more dampened signal of the soil moisture readings.



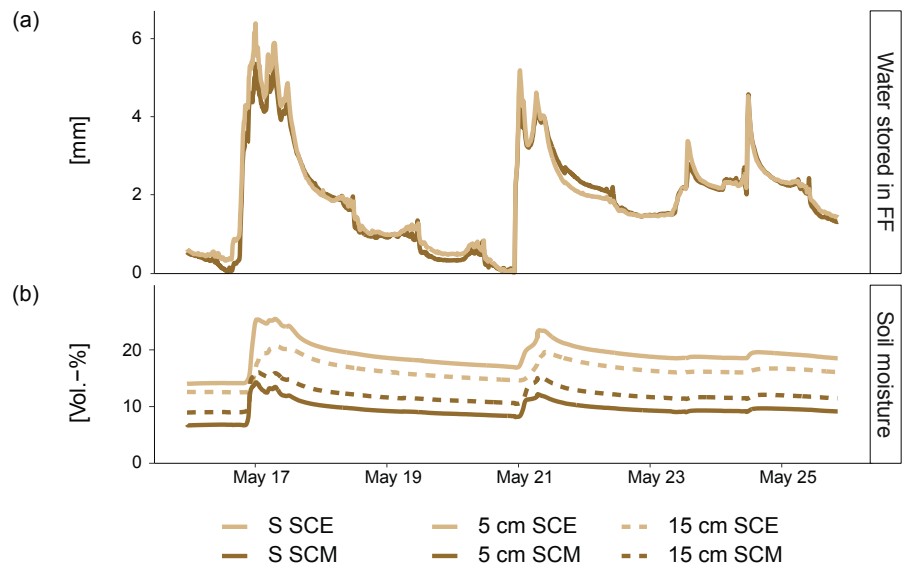

| S SCE | 5 cm SCE | 15 cm SCE |
| S SCM | 5 cm SCM | 15 cm SCM |


**Figure 8: (a) Water stored in the FF quantified by the lysimeter and (b) soil moisture content next to the lysimeters.**

## 4 Discussion

The lysimeter data proves to be a very valuable addition to existing approaches such as grab sampling or rainfall simulator experiments (Li et al., 2020; Putuhena and Cordery, 1996; Walsh and Voigt, 1977; Zhao et al., 2022). It includes environmental

factors like climatic conditions, pre-wetness, canopy structure, litter composition, soil structure, etc. Evaporation, retention, and storage processes are assessed in actual field conditions. Still, data needs quality/plausibility checks and validation by supplementary measurements or data from the site. For example, during long term observation it is important to consider the weight loss of the decomposing FF material. Weight increases may result from the relocation of material caused by falling branches, leaves, and other debris being moved in and out of the box by wind. The typical factors of these lysimeters have

been previously described by Gerrits et al. (2007). They state that many factors can influence the accuracy and errors of a forest floor lysimeter, like evaporation during a rainfall event, measuring noise, falling branches/leaves, small animals, dew and heterogeneity of throughfall. Another disadvantage of lysimeters is that only one specific surface and volume is taken into account, which might not cover the overall heterogeneity at different spatial scales (von Unold and Fank, 2008). We installed multiple lysimeters for direct comparison among the FFGLs at each site and to validate with above-canopy precipitation. So,

we can make sure that a sudden storage change might not be caused by precipitation but rather by some form of disturbance. Even smaller spatial heterogeneity, with a spatial scale of 0.0125 m², will be displayed through the grids of the lysimeter. Our



initial data indicate that even on a small scale, there are significant differences in drainage, with one grid exhibiting over 50% drainage and another only 5%.

Usually lysimeters are very costly and extensive to construct and install. Bello and Van Rensburg (2017) conducted a review
of small lysimeter costs, which ranged from US$ 1,000 to US$ 5,500. Initial efforts are being made to create low-cost lysimeters. Bello and Van Rensburg (2017) built a lysimeter for US$ 520 and Dong and Hansen (2023) built a lysimeter station including four separate lysimeters for US$ 1,310. Due to specific requirements for our lysimeter, including grids for spatial heterogeneity, shallow depth to cover only the upper field capacity, and adaptability for steep research sites, we opted to explore a DIY low-cost setup, as customized lysimeters are typically costly. The manufacturing price of our FFGL is currently
US$ 750, with the stainless-steel container accounting for US$ 550 of this cost. Utilising an alternative material for the container could significantly reduce costs.

The results presented show that water flux measurements with the FFGL are very precise. We achieve a resolution of 0.03 mm for precipitation detection with a SD of 0.0048 mm and an error of 2%. This is comparable to other lysimeter studies. Bello and Van Rensburg (2017) developed a small lysimeter with a resolution of 0.123 mm, Dong and Hansen (2023) achieved a
resolution of 0.3 mm and Ruth et al. (2018) used lysimeters with a resolution of approximately 0.016 mm and a Mini-Lysimeter with a resolution of 0.5 g, equivalent to a water column of approximately 0.007 mm. Seuntjens et al. (2001) conducted a comparison of the precision across various lysimeter studies, revealing that the precision for lysimeters with a cross-sectional area less than 1 m² varied between 0.025 mm and 0.5 mm.

To enhance future measurements, we will include a specific calibration for each TB and not use a mean tipping volume for all.
Segovia et al. (2021) discuss issues related to tipping bucket results, including heterogeneity in bucket manufacturing, the initial wetness or dryness of the bucket, and the tension between the reed switch and magnet. Furthermore, we will incorporate temperature corrected LC measurements, since we realized that some of our LCs show a temperature dependency despite the information of the manufacturer that the LCs are temperature compensated. This makes field measurements more challanging since we need to correct for temperature variations. But these adjustments will help to improve measurements and also make
it possible to precisely measure evaporation as well as to detect non-rainfall precipitation such as dew and rime.

## 5 Conclusion and Outlook

The FFGL underwent successful field testing and will now be installed at various sites and in different positions relative to trees. The objective is to gather data on the effects of climatic conditions, tree species, crown position, and forest floor (FF) composition on the water balance of the FF. Additionally, we aim to investigate the variability of infiltration patterns resulting
from throughfall patterns and FF composition. The tested setup facilitates the future integration of water quality measurements into the existing measurement units. The development of automated in situ measurements for water temperature, electrical conductivity (EC), and dissolved organic carbon (DOC) in the grid-lysimeter is currently underway.





**Code availability**

https://github.com/HeinkePaulsen/Forest-Floor-Grid-Lysimeter

**Data availability**

Forstliche Versuchs- und Forschungsanstalt, Abt. Boden und Umwelt (Forest Research Institute Baden-Württemberg, Dept. Soil and Environment)

https://github.com/HeinkePaulsen/Forest-Floor-Grid-Lysimeter

**Author contribution**

MW and HP designed the setup. HP developed the Lysimeter setup. HP prepared the manuscript with contributions from all co-authors.

**Competing interests**

At least one of the (co-)authors is a member of the editorial board of Hydrology and Earth System Sciences.

**Acknowledgements**

*This project was carried out in the framework of the Research unit 5315 "*Forest Floor: Functioning, Dynamics, and Vulnerability in a Changing World*" funded by DFG.*

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
