# Peer review of "Technical note: a Weighing Forest Floor Grid-Lysimeter"

_EGUsphere, 2024_

## Author Response (AR1)

Responses to editor and reviewer comments for

**"Technical note: a Weighing Forest Floor Grid-Lysimeter"**

Responses in blue Added text in purple

**Response to the Editor**

Dear authors,

As can be seen both reviewers have minor comments on your manuscript and I agree with your proposed revisions. A few additional (minor) comments from my side which might help to improve the manuscript further:

Dear Dr. Coenders-Gerrits,

Thank you for handling our manuscript and for the opportunity to submit a revised version. Below we provide point-by-point replies to all comments made by you the editor and the reviewers.

With kind regards,

Heinke Paulsen (on behalf of all co-authors)

Explain what the added value is of the 4 compartments? In addition, you present that the smaller surface area (0.25m2) of your device in comparison to others (±1m2) is beneficial. However, I tend to disagree as 'edge effects' (water flowing along the edges) become more dominant. Maybe briefly discuss

**Reply** Due to the 4 compartments we are able to measure the redistribution of precipitation water in the FF layer. When incoming precipitation is evenly distributed (which becomes evident in similar weight changes of the single LCs) but the TBs show diverging portions of drainage, the FF takes a big part in this redistribution. We added this point to the discussion (Sec. 4). Special for our device is that the FF material isn't partitioned by walls, which we tried to clarify in Sec. 2.1. Therefore, possible edge effects can only happen along the main wall of the lysimeter. The overall size of 0.25 $m^2$ was selected in order to install the lysimeter in different sloped terrain. A 1 $m^2$ would not be possible, so we decided on a compromise. Additionally, our goal was to have an easy "DIY" setup that could be handled for maintenance by maximum two people. In case other studies using the lysimeter would like to increase the size, this is easily possible.

Sec. 2.1: On top of these dividing walls lies a perforated metal sheet wrapped with geotextile to retain the forest floor material, which is placed on top. Like this there is no disruption of the FF and we can potentially show redistribution of water induced by the FF layer.

Sec. 4: When incoming precipitation is evenly distributed over the lysimeter area, which becomes evident in similar weight changes of the single LCs, diverging portions of drainage in the TBs are a sign of redistribution by the FF.

Sec. 4: It needs to be taken into account that lysimeters with a smaller surface area show more dominant edge effects. The overall size of 0.25 m$^2$ in our setup was selected in order to install the lysimeter in different sloped terrain, so we decided on a compromise. Additionally, our goal was to develop a low-cost setup that wasn't too big in size and weight and could still be handled by maximum two people for maintenance.

You discuss the accuracy of your device. However, I think your device consists of two accuracy: the loadcells of the top basin (to determine forest floor interception/storage) and the tipping buckets (to measure infiltration)

**Reply** We adapted this part in the discussion.

We achieve a total resolution of 0.03 mm for precipitation detection with a SD of 0.0048 mm and an error of 2%, when we combine the measurements by the LCs and the TBs. Small precipitation (rime, fog, dew) that doesn't produce drainage from the lysimeter and evaporation can be measured with a resolution of 0.008 mm.

L198-200: please remove the links to the commercial (reseller) websites (who are even not from the manufacturer)

**Reply:** We removed them.

Eq 2+3: please add the time component here. Either by dividing dS by dt, or by using integrals/summation

**Reply:** We changed the equations by adding the time component.

**Response to Reviewer #1**

I enjoyed this simple paper describing an innovative way to measure fine-scale fluxes of water in the forest floor. The presentation was generally clear and I can find no fault with the description of the device or the demonstration of its capabilities. I have some suggestions to improve the presentation in a few instances.

**Reply** We appreciate the helpful comments of the Anonymous Referee #1. We implemented all technical corrections and the comments are answered below.

"weighted" lysimeter appears several times where "weighing" lysimeter is meant.

**Reply** We changed this in the revised paper.

L84 what is an SDI-12 sensor?

**Reply** SDI-12 sensors are common to be used in environmental applications. They use a common digital protocol for interconnection of sensors and dataloggers and are characterized by a low energy consumption. We added a short explanation.

In environmental applications the use of SDI-12 sensors is very common, since they use a common digital protocol for interconnection of sensors and dataloggers and are characterized by low energy consumption. This is why we incorporated this feature into our board to enable the use of the FFGL as an SDI-12 sensor.

Sec 2.2.2 many of these comparisons to alternatives are not very helpful, given that they mainly refer to old hardware. An eye to making the description of the device more timeless would improve its long-term utility.

**Reply** In the revised version, we put more focus on our board, instead of the comparison to older hardware.

Sec 2.3 I think the precision and resolution of the load cells is more important than accuracy. Please explain how precisely changes in mass can be resolved. Also I am curious about temperature dependence.

**Reply** In Sec 3.1 we show, that mass changes of 2 g corresponding to 0.008 mm of precipitation can be detected. In the revised version of the manuscript we added a reference in Sec 2.3 to the results (Sec 3.1) and that numbers for the resolution will be shown there.

By now we found that the temperature dependence addressed in Sec 4 does not derive from the load cells themselves but was caused by inadequate positioning of the lysimeter box and therefore suboptimal force application to the load cell. This caused "weight changes" when the stainless steel box shifted due to thermal expansion. We changed the paragraph accordingly.

Since we realized a temperature effect in our field weight measurements we performed some temperature and stability tests of our LCs. The change in weight did not occur then, showing the effect in our data was caused by shifting of the lysimeter container due to thermal expansion and suboptimal force application to the LC due to wrong positioning of the box. For future measurements we adapted the setup.

L145 a narrow hole may prevent calibration problems with the bucket, but it does degrade the ability of the device to precisely estimate time of infiltration. That is worth mentioning, along with any information you might have to quantify this effect.

**Reply** Since we do not detect the timestamp of every single tip but aggregate the number of tips over the measurement interval (e.g. 10 minutes) we already degraded this very precise estimate of infiltration. Usually in this interval all the water will pass the tipping bucket. But of course, this is worth mentioning. We added a number of how much water could flow through the MU in an interval of 10 minutes.

Due to the aggregation of tips during the measurement interval we neglect the degradation in precision by this restriction. The outflow still allows for a correct measurement in a 10-minute interval of 3 liters which would be equivalent to 48 mm water depth, which is typically much higher than extreme 10-minute precipitation events.

L150 can you give some indication of the morphology of the forest floor? Is this a mor humus with a distinct organic-mineral transition, or a mull humus where the transition is indistinct? I don't think this matters much for the description of the device, but it does give some context on what is possible using this measurement technique.

**Reply** It's a typical Moder. We added this information in Sec. 2.5 and 3.4.

L152 what is garden fleece?

**Reply** A water permeable geotextile that allows water to pass but keeps soil and other particles from draining from the box. We changed the wording in the revised manuscript.

Figure 3 would improve if it included indications of the reference masses

**Reply** We added this information to the figure caption.

L214 I don't understand this sentence. It sounds like the second thing is a consequence of the first, and that they are not two separate effects.

**Reply** We separated this sentence to make clear that the retention of infiltration and amount of stored water are two separate points.

For the second and third event the drainage occurred much faster (3 minutes, with 0.5 & 0.2 mm water applied) due to the higher initial water content. Also, during these two events a larger part of the irrigated water drained from the box.

L218 I don't understand this sentence. Perhaps fumigation is the wrong word?

**Reply** We changed the wording.

Figures 4 and 5 would improve if the width of the time bins was specified.

**Reply** We added the specification in the figure captions.

Instead of SCE and SCM, which are difficult to remember, why not "crown edge" and "crown middle" in all figures? There is plenty of room available.

**Reply** We changed the labelling in all figures.

Table 2 what is 16.5.-26.5.?

**Reply** It's the date of the 10-day period, we removed it.

Throughout the results: in a few places there are explanations for various things like why percept throughfall varied (L241, L267), but these are not reliable conclusions due to low sample size.

**Reply** Of course the presented results in this technical paper are not enough to draw strong conclusions on the processes going on in the forest floor. But we show them to illustrate what conclusions **could** be drawn from lysimeter data with a higher sample size in the future and that the FFGL is a valuable addition to existing approaches. We changed the sentences to be more hypothetical.

**Response to Reviewer #2**

The technical note "A Weighing Forest Floor Grid-Lysimeter" by Paulsen and Weiler reports the technical details, performance tests and preliminary analyses of a novel lysimeter that allows estimating the water storage and retention capacity of the forest floor. The technical innovation is highly appreciated, a lot of work must have been gone in the development of the lysimeters. However, the manuscript sometimes is a little hard to follow i.e., when it's not clear what pointers refer to, some sentences could be rephrased to make them shorter and clearer. I'd like to suggest some edits that might help to improve the manuscript, but please thoroughly edit the entire manuscript to improve the overall readability. Figure captions are often very short and lack important information, maybe you can improve them during revision.

**Reply** We appreciate the helpful comments of the Anonymous Referee #2. All technical corrections were implemented and the comments are answered below. We improved the readability during revision and also further clarified and extended the figure captions.

Overall I only identified one critical aspect that is related to the accuracy of load cells. Most (cheaper) load cells suffer from uncertainty with changing temperatures (as the authors also acknowledge) and additionally they have weak long-term stability in the measured weight (drift). Did you do any long-term reliability testing (ideally with different temperatures) and could you show these results (e.g., in the supplement)? I understand that new measurements and tests might not be an option, so therefore I'd suggest to at least discuss this topic thoroughly and report the metrics provided by the load cell supplier in the manuscript. Maybe a paragraph can be added in the discussion.

**Reply** As mentioned in the Response to Referee #1 by now we found that the temperature dependence addressed in Sec 4 does not derive from the load cells themselves but was caused by inadequate positioning of the lysimeter box and therefore suboptimal force application to the load cell. This caused apparent "weight changes" when the stainless-steel box shifted due to thermal expansion. We performed additional tests regarding long term reliability (also under different temperatures) and show them in the supplement. For our field setup we repeatedly place a known weight on the lysimeters during maintenance to test the long-term stability in the field Also, we changed the part in Sec 4, explaining the phenomenon.

Since we realized a temperature effect in our field weight measurements we performed some temperature and stability tests of our LCs. The change in weight did not occur then, showing the effect in our data was caused by shifting of the lysimeter container due to thermal expansion and suboptimal force application to the LC due to wrong positioning of the box. For future measurements we adapted the setup. Additionally, it is necessary to test the long-term stability of load cells. We do this by repeatedly placing a known weight on the lysimeters during maintenance to test the long-term stability in the field.

L17: induce runoff generation? Should this be "reduce"?

**Reply** Yes, we changed it.

L 19-20: rephrase the sentence: I do not think that these are "ecosystem services" of canopies, but you rather want to state that similar processes occur in canopies & FF, also I'd rephrase the processes: e.g., retention (is important but missing currently), water redistribution (I think this is meant by water infiltration) - > maybe more general: canopies and FF affect the temporal, spatial distribution but also chemical composition of soil water recharge.

**Reply** This is a good objection. We adapted the phrase.

L 39: not clear what "true" precipitation is

**Reply** It's the exact amount of water reaching the surface. Other measurements like rain gauges are affected by e.g. wind and therefore have a higher error. We clarified what we mean by "true".

L 65: remove: "without further securing measures" -> not clear what it means to me.

**Reply** We removed it.

L84: rephrase: the output data from the FFGL are in SDI-12

**Reply** We rephrased the sentence according to Referee #1 to make it more understandable.

Figure 4: water flux in mm / per time, write out precipitation and drainage in the legend, better call it "cumulative water amounts" & Figure 5: mention in the caption which lysimeter is below the edge (SCE) vs. the middle of the canopy (SCM) &. Figure 7: make clear what the two lysimeters are (i.e., mention below the edge (SCE) vs. the middle of the canopy (SCM)) & Figure 8: also here the Figure caption lacks some information

**Reply** We changed the labelling in the figures, reduced the abbreviations and made more clear captions.

Table 2: it would be good to reiterate the abbreviations in the caption

**Reply** We changed that.

Figure 6, caption: why are some words capitalized? better call it "cumulative water amounts", not clear what "measured on the tower" means, say above the canopy?

**Reply** We changed the figure caption.

L 271: could & could, maybe this sentence can be rephrased

**Reply** We rephrased the sentence.

One could state that this occurs due to spatial heterogeneity in $P_{TF}$, but since the mass gain on all four LCs was similar, we conclude that this deviation in infiltration must be due to rerouting pathways in the FF itself.

L 277: how is this 5mm to 10% conversion done?

**Reply** When the amount of stored water rises from 0 to 5 mm the soil moisture rises from 15% to 25%.

L 285 – 287: The sentence doesn't make sense to me. i.e., I'd start with "The data include…" but it's not clear how the lysimeter data include information on climate conditions, canopy structure; "e"vaporation -> now I understand, etc. misses a second dot, still not very clear.

**Reply** We rephrased the sentence.

Evaporation, retention, and storage processes are assessed in actual field conditions, while environmental factors like climatic conditions, pre-wetness, canopy structure, litter composition, soil structure, etc. stay incorporated.

L 314: not clear what that means

**Reply** We rephrased this sentence

To enhance future measurements, we will include specific calibrations for each TB and not use one mean tipping volume for all TBs.

The "conclusions & outlook" are only an outlook, maybe you can rewrite this

**Reply** We added a paragraph to the conclusion.

This technical note describes the development of the low-cost weighing Forest Floor Grid Lysimeter (FFGL). We present accuracy tests for the different water flux quantifications, infiltration measured with the 3D-printed tipping buckets and storage weight measurements, resulting in a total resolution of 0.03 mm for precipitation detection with a SD of 0.0048 mm and an error of 2%. The separately draining grids without walls between the compartments dividing the FF allow for quantification of redistribution induced by the FF.